# Photobiomodulation of Neurogenesis through the Enhancement of Stem Cell and Neural Progenitor Differentiation in the Central and Peripheral Nervous Systems

**DOI:** 10.3390/ijms242015427

**Published:** 2023-10-21

**Authors:** So-Young Chang, Min Young Lee

**Affiliations:** 1Beckman Laser Institute Korea, Dankook University, Cheonan 31116, Republic of Korea; so4040@hanmail.net; 2Department of Otolaryngology-Head &Neck Surgery, College of Medicine, Dankook University, Cheonan 31116, Republic of Korea

**Keywords:** photobiomodulation, neurogenesis, stem cell, nervous systems

## Abstract

Photobiomodulation (PBM) is the regulation of biological processes using light energy from sources such as lasers or light-emitting diodes. Components of the nervous system, such as the brain and peripheral nerves, are important candidate PBM targets due to the lack of therapeutic modalities for the complete cure of neurological diseases. PBM can be applied either to regenerate damaged organs or to prevent or reduce damage caused by disease. Although recent findings have suggested that neural cells can be regenerated, which contradicts our previous understanding, neural structures are still thought to have weaker regenerative capacity than other systems. Therefore, enhancing the regenerative capacity of the nervous system would aid the future development of therapeutics for neural degeneration. PBM has been shown to enhance cell differentiation from stem or progenitor cells to near-target or target cells. In this review, we have reviewed research on the effects of PBM on neurogenesis in the central nervous system (e.g., animal brains) and the peripheral nervous system (e.g., peripheral sensory neural structures) and sought its potential as a therapeutic tool for intractable neural degenerative disorders.

## 1. Introduction

Neurological diseases such as Alzheimer’s disease, Parkinson’s disease, stroke, multiple sclerosis, epilepsy, migraine, myasthenia gravis, Huntington’s disease, amyotrophic lateral sclerosis (ALS), and peripheral neuropathy affect the nervous system, which includes the brain, spinal cord, and nerves throughout the body [1,2,3,4,5,6,7]. These conditions vary widely in terms of symptoms and severity, and they can be caused by a variety of factors including genetics, infections, immune system disorders, trauma, and environmental factors [8,9,10,11].

The two main nervous systems are the central nervous system (CNS) and peripheral nervous system (PNS), which have distinct anatomical locations and functions within the body [12]. The CNS includes the brain and the spinal cord; it is the central processing unit of the nervous system and the main region where the integration and coordination of sensory and motor information occurs [13]. The PNS includes all nerves and ganglia (collections of nerve cell bodies) that lie outside the CNS; it connects the CNS to the rest of the body, including muscles, organs, and sensory receptors. Thus, the PNS transmits sensory information, including touch, temperature, pain, and other sensory stimuli, from the body’s periphery (e.g., skin, muscles, and organs) to the CNS, which then interprets the signals and formulates appropriate responses [14,15,16].

Neurological diseases are progressive, i.e., they worsen over time. Although many studies have contributed to the development of therapeutic methods to treat such diseases, their precise mechanisms remain unknown, and there is a lack of specific indicators, such as biomarkers, to aid diagnosis and treatment. By the time neurological symptoms become apparent, serious damage may already have occurred, making it difficult to reverse their effects.

Photobiomodulation (PBM), also known as low-level light therapy, is a non-invasive technique that uses lasers or light-emitting diodes (LEDs) with low-level light in the visible and near-infrared (NIR) spectra to regulate biological processes [17,18]. PBM research is being conducted worldwide and has already been shown to be effective in clinical applications, such as tissue repair or pain relief, and in various medical applications including sports medicine [19,20,21,22,23,24,25]. PBM is thought to enhance cellular metabolism and other processes, eventually leading to increased cell proliferation and differentiation [26]. Nervous system components such as the brain and peripheral nerves are important candidate PBM targets due to the lack of current therapeutic modalities for the complete cure of neurological diseases. There are two approaches to the application of PBM in neurological diseases: the prevention or reduction in damage caused by the disease and the regeneration of damaged organs. Recent findings suggest that neural cells can regenerate, which contradicts our previous understanding of these cells. Neurogenesis is the process by which new neurons are created from neural stem cells and progenitor cells. Although this process was once thought to be limited to early development, it continues to occur in two specific regions of the adult brain: the hippocampus and the olfactory bulb [27,28,29,30]. Nevertheless, neural structures are still thought to have weak regenerative capacity compared to other systems [31,32,33,34]. Therefore, enhancing the regenerative capacity of the nervous system could aid the future development of therapeutics for neural degeneration. PBM has been shown to enhance cell differentiation from stem or progenitor cells to near-target or target cells. An overview of the potential targets for PBM-based neurogenesis is shown in Figure 1. Additional details about the papers depicted in Figure 1 can be found in Table 1 and Table 2.

In this review, we have reviewed research on the effects of PBM on neurogenesis in the CNS (e.g., animal brains) and the PNS (e.g., peripheral sensory neural structures) and explored its potential as a therapeutic tool for neural degenerative disorders whose cures are intractable with current therapeutic approaches.

## 2. Basic Mechanisms of PBM for Cell Differentiation

PBM uses light in the red or NIR spectra to stimulate cellular processes and promote various biological effects. Therefore, to effectively harness the potential of PBM for influencing and regulating cell differentiation, a methodology to maximize light parameters, chromophores, and mitochondrial activity is required. 

Light parameters such as wavelength, power density, energy density, and treatment duration are important factors that determine the effectiveness of PBM. PBM is mainly biostimulated through light absorption by the chromophore cytochrome c (Cytc) oxidase; therefore, the range of wavelengths that can be absorbed by Cytc oxidase limits PBM activation [51]. Thus, the most commonly used wavelengths for PBM activation are in the red (600–700 nm) and NIR (700–1100 nm) regions, as these wavelengths penetrate deeper into tissues and are more effective in stimulating cellular responses [26,52]. The absorption of light energy by chromophores such as Cytc oxidase, a key enzyme involved in cellular respiration, triggers a series of biochemical reactions and cell signaling pathways that play important roles in cell differentiation. In particular, mitochondria are directly affected by the action of Cytc oxidase. 

The stimulation of mitochondrial activity is among the primary mechanisms of PBM. Mitochondrial stimulation by absorbed light energy enhances the production of adenosine triphosphate (ATP), which is a major energy source for cellular processes. Increased ATP production boosts cell metabolism and provides energy for cell differentiation [53]. PBM promotes the recovery of damaged nerve cells through mitochondrial fission imbalance [54] and influences various intracellular signaling pathways involved in cell differentiation. It modulates the activity of transcription factors, such as cyclic adenosine monophosphate, mitogen-activated protein kinases, nuclear factor kappa B, and neuroligin-3, which play vital roles in regulating gene expression and determining cell fate [18,55,56,57]. The intracellular stimulation of a specific wavelength by PBM affects gene expression and stem cell differentiation and is involved in cell differentiation. PBM can affect gene expression patterns, leading to the up- or downregulation of specific genes involved in cell differentiation. It also promotes the expression of genes related to differentiation markers and suppresses genes associated with proliferation, thereby favoring differentiation processes. PBM has shown the potential to enhance the differentiation of stem cells into specific neural lineages. For example, PBM activated by NIR wavelengths increased the differentiation of adipose stem cells immortalized by human telomerase enzymes into neurons, and PBM treatment using growth factors such as basic fibroblast growth factor promoted the differentiation of adipose stem cells to neuron cells [58,59].

## 3. Promising Results

### 3.1. Effects of PBM in the CNS

#### 3.1.1. Neurogenesis

PBM has been investigated for its potential effects on neurogenesis in the CNS, particularly those involving stem cells and neural progenitor cells. PBM has been shown to stimulate the proliferation of neural stem cells and neural progenitor cells into neurons. PBM strongly inhibits reactive gliosis and promotes inflammatory cytokine production, and it can enhance mitochondrial function to improve the neuronal microenvironment, ultimately affecting nerve regeneration [48]. Furthermore, it influences the expression of genes and signaling pathways involved in neuronal differentiation, thereby facilitating the transformation of progenitor cells into functional neurons.

The hippocampus, a brain region important for learning and memory, has been investigated as a region that may influence the effects of PBM on neurogenesis in the CNS. One study found that PBM increased the survival of neonatal neurons in the hippocampus by activating the AKT/GSK-3β/β-catenin pathway [35,36]. Transcranial PBM appears to improve learning, memory, and neural progenitor cell function after traumatic brain injury in rodents; PBM irradiation for 7 consecutive days after photothrombosis induction in an ischemic stroke rat model reduced the infarct area and increased the expression of the neurogenesis and synaptic markers bromodeoxyuridine (BrdU), Ki67, doublecortin, MAP2, spinophilin, and synaptophysin. Moreover, NIR-PBM can reduce cell death and improve learning and memory performance by increasing the expression of neuroprogenitor cells, doublecortin, and TUJ1 after controlled cortical impact or traumatic brain injury induction in mice [37]. Chronic PBM can promote adult hippocampal neurogenesis by activating latent transforming growth factor β1 (TGFβ1) [38]. The papers on PBM and neurogenesis mentioned in the content can be summarized as in Table 1. PBM also enhances the migration and differentiation of neural stem cells in the subventricular zone within the walls of the lateral ventricle, which is another brain region involved in neurogenesis [60].

#### 3.1.2. Neuroprotective Effects of PBM

PBM has neuroprotective effects that support the survival of newly generated neurons during neurogenesis [61,62]. These effects include increasing the production of anti-apoptotic proteins, enhancing cell survival pathways, and reducing oxidative stress and inflammation, thereby creating a favorable environment for the survival and integration of new neurons [39,40,49]. These effects are induced by the activation of AKT, which suppresses the activity of glycogen synthase kinase 3β (GSK3β) and the activation of the extracellular signal-related kinase/forkhead box protein M1 pathway [63]. The effects of PBM are not limited to stress reduction; they also include other neuronal functions such as homeostasis, which is maintained by microglial cells, as well as nerve fiber growth and enhanced synaptic connections between neurons to promote their maturation and differentiation [41,42,43].

##### Glial Cells

PBM affects glial cells including microglia, astrocytes, and oligodendrocytes, which support neuronal structures. PBM may promote nerve recovery following ischemic stroke by converting the M1 microglial phenotype into the anti-inflammatory M2 phenotype [48] or reducing the expression of tumor necrosis factor-α, interleukin (IL)-1β, and IL-6, which are involved in pro-inflammatory responses in the M1 or M2 microglial phenotype [50]. A combination of PBM and a static magnetic field exerted neuroprotective effects by reducing GFAP and Iba1 expression, oxidative stress, and apoptosis, following their increase through the injection of Aβ_25-35_ peptide, which causes Alzheimer’s disease; overall, the treatment appeared to improve memory [44]. PBM also influences astrocyte migration and protects astrocytes by maintaining astrocyte proliferation or differentiation; the increased co-expression of GFAP and BrdU/Ki67 indirectly supported this theory [64]. PBM also attenuated oligodendrocyte dysfunction and prevented adverse neurological consequences in a rat model of early life adversity [65].

##### Synaptogenesis

PBM has the potential to facilitate synaptogenesis, which is the formation of synapses between neurons. It can enhance the development of dendritic spines, promote axonal growth, and facilitate the establishment of functional connections between newly generated neurons and existing neural circuits [45]. After applying PBM to a temporal lobe epilepsy model, neurodegeneration and cognitive decline were attenuated by increasing the expression of the synapse-related protein neuroligin-3 [57]. PBM has also been suggested to promote functional recovery after stroke by inhibiting the neurotoxic astrocyte marker C3d, enhancing synaptic expression, and regulating endogenous apoptotic pathways [46]. PBM via NIR irradiation alleviated anxiety and depression symptoms, as well as neuronal cell death, in TgF344-AD mice, and it suppressed neuronal degeneration by increasing the expression of MBP, MAP2, and PSD95, which are nerve fiber and synapse markers [47]. The papers on PBM and neuroprotective effects, glial cells, and synaptogenesis mentioned in the content can be summarized as in Table 2.

After the regeneration of the soma, cell-to-cell connections are necessary for neuronal structural function and synaptogenesis; neural network formation is crucial in forming these connections. PBM via long-term transcranial NIR irradiation was found to improve excitatory neurotransmission and oxidative metabolism in young rats and to alter metabolic pathways in aged rats to levels found in young rats through the increased survival and differentiation of neonatal neurons [66]. This excitatory neurotransmission may be augmented by stronger synaptic connections.

### 3.2. Effects of PBM on Peripheral Sensory Neural Structures 

#### 3.2.1. Peripheral Neural Diseases

Peripheral neural diseases, particularly sensory neuronal diseases, have various underlying causes [67,68]. For example, hearing loss may be a symptom of peripheral neuropathy, multiple sclerosis, Charcot–Marie–Tooth disease, or Friedreich’s ataxia [68,69,70,71]. Peripheral neuropathy is among the most common peripheral neural diseases causing hearing loss [72]. Hyposmia, or a decrease in the sense of smell, is a symptom of idiopathic olfactory loss, idiopathic olfactory dysfunction, and Guillain–Barré syndrome [73,74,75]. Guillain–Barré syndrome is an autoimmune disorder that affects the PNS and can involve the peripheral nerves responsible for taste [76]. Ramsay Hunt syndrome can impair the function of multiple sensory organs, causing severe sensorineural hearing loss and sometimes leading to olfactory nerve damage that causes a loss of smell. A characteristic symptom of Ramsay Hunt syndrome is facial paralysis caused by damage to the facial nerve, which can also lead to the loss of taste [77]. 

Conditions that result in the loss of hearing, smell, and/or taste have a major impact on quality of life. Deafness can result in communication difficulties, which can lead to feelings of isolation and frustration during social interactions. Sensory impairments, such as the loss of smell and taste, and facial nerve disorders can limit normal social functioning by reducing emotional well-being and the quality of interpersonal relationships [78,79]. In addition to affecting quality of life through loss of the senses, peripheral nervous disease eventually leads to CNS degeneration. Therefore, recovery of these senses is an important target of current PBM research. Ongoing trials are applying pluripotent cells, such as stem or progenitor cells, to regenerate sensory neural structures. However, these trials have been unsuccessful to date, showing poor survival and a lack of cell differentiation. Therefore, few published studies have demonstrated stem cell PBM augmentation in anosmia and other sensory disorders. 

#### 3.2.2. Hearing Loss, Stem Cells, and PBM

The main cause of hearing loss is damage to cochlear tissue in the inner ear. Normally, hair cells in the cochlea convert external mechanical signals into electrical signals and transmit them to the brain. Hair cell damage disrupts this process, causing hearing loss. Because hair cell damage or loss is irreversible, it is very difficult to treat hearing loss. Therefore, the rescue of hair cells is an important target for therapeutic treatment. Sensorineural hearing loss treatment using stem cells is currently being investigated, with a focus on restoring damaged hair cells. However, due to the structural complexity of the inner ear and the characteristics of the environment within the cochlea, the engraftment and survival rates of stem cells transplanted from outside the body have been insufficient to restore hearing loss. To overcome these limitations, some studies have attempted a PBM-based promotion of cell differentiation for cochlear hair cell regeneration. In an auditory study, otic cell differentiation was enhanced through otic vesicle formation and an increased proliferation of myosin VIIa-positive cells after three-dimensional mESC cultures were irradiated with 630 nm LED light to induce PBM. [80]. Non-invasive irradiation (>800 nm) of the inner ear through the tympanic membrane strengthened otic-like organoids and enhanced the adhesion and differentiation potential of transplanted stem cells within the cochlea [81]. The development of hearing rehabilitation technologies, such as cochlear implants, has allowed patients with profound hearing loss to regain their hearing [82]. However, spiral ganglion neurons, which are auditory neurons that transfer neural signals from the cochlea to the brainstem, must be healthy and functional to attain optimal cochlear implant performance [83]. Considering that these auditory neurons degenerate following hearing deprivation, the performance of cochlear implants tends to decline over time. Thus, the discovery of a technique for regenerating spiral ganglion neurons from stem cells would be highly beneficial to those with hearing loss. Although several studies have proposed methods for the differentiation of auditory neurons from stem cells [84,85,86], optimal methods have not been demonstrated for such auditory neuron differentiation within the cochlea or for stem cell migration to Rosenthal’s canal, where auditory neurons reside. PBM may not be best tool for inducing stem cell differentiation in Rosenthal’s canal because it is located in the medial section of the cochlea. Considering the low penetration rate in this anatomical region, PBM efficiency would be extremely low. However, there are alternatives, such as soluble hydrogel recently developed for delivering light energy to remote anatomical locations [87]. This hydrogel can be modified to encapsulate cell differentiation and homing factors for the neural differentiation of stem cells. Such an approach could allow us to selectively activate stem cells using light energy. A schematic of this methodology is shown in Figure 2. 

## 4. Conclusions and Future Directions

PBM has shown potential to promote neurogenesis in the CNS or PNS by enhancing cell differentiation from stem cells or neural progenitor cells to nearby target cells or target cells. Nevertheless, several limitations and challenges must be considered. As PBM employs non-invasive light, the tissue penetration rate varies depending on the light source utilized, and the results may exhibit biphasic patterns depending on the frequency of irradiations and energy dosage. In other words, an inappropriate optimal dose may either yield no PBM effect on the target or be excessively high, suppressing the intended PBM effect [52,88,89]. Consequently, PBM can be applied to various areas of the body, but to elicit positive treatment outcomes, it is imperative to establish an appropriate protocol tailored to the target tissue. However, because the PBM parameters presented in the papers reported so far are very diverse, the standardization of PBM has not been clearly established to date. Moreover, ongoing research into PBM’s capacity to stimulate neurogenesis in both the central and peripheral nervous systems underscores the evolving nature of this field. Therefore, while the field continues to progress, it is important to consider potential safety and side effects.

Determining the optimal PBM parameters for neurogenesis remains challenging. Parameters such as the wavelength, energy density, treatment duration, and timing of PBM application require further optimization. Different cell types and experimental conditions may require different parameters, making it difficult to establish a standardized protocol. Stem and neural progenitor cell populations in the CNS or peripheral sensory nerves are heterogeneous. Different subtypes of stem or progenitor cells may respond differently to PBM, and their specific requirements for differentiation and neurogenesis may vary. Future advances will depend on understanding this heterogeneity and tailoring PBM approaches accordingly.

Light penetration through tissues is limited, particularly in the CNS. The depth of penetration depends on the light wavelength. Although NIR light can penetrate deeper into tissues than visible light, reaching target cells deep within the CNS may remain a challenge. This limitation could impact the effects of PBM on neurogenesis in deep regions of the CNS, complicating the precise targeting of stem cells or neural progenitor cells for PBM. Ensuring that light reaches the desired cell population within complex neural tissue structures is also difficult. Strategies such as the application of cell-specific markers or targeted delivery methods must be further explored to enhance the specificity of PBM.

The response to PBM can vary among individuals and experimental conditions. Factors such as age, health status, genetic background, and diseases/conditions can influence the cellular response to PBM. To draw reliable conclusions, it is important to standardize experimental conditions and consider variability in cell responses. The long-term effects and safety profile of PBM for neurogenesis require further investigation. PBM has shown promising results in preclinical studies; however, further research is needed to assess its potential adverse effects, determine optimal treatment regimens, and ensure the long-term safety of PBM-based approaches. Thus, although PBM research has demonstrated benefits in preclinical models, transforming these findings into effective clinical applications is a complex process. Clinical studies involving human subjects are necessary to determine the efficacy, safety, and feasibility of PBM for neurogenesis in the CNS or PNS. Addressing these limitations through further research, optimizing PBM parameters, and designing appropriate clinical trials will contribute to a better understanding of the potential of PBM for neurogenesis and its application in regenerative medicine.

## Figures and Tables

**Figure 1 ijms-24-15427-f001:**
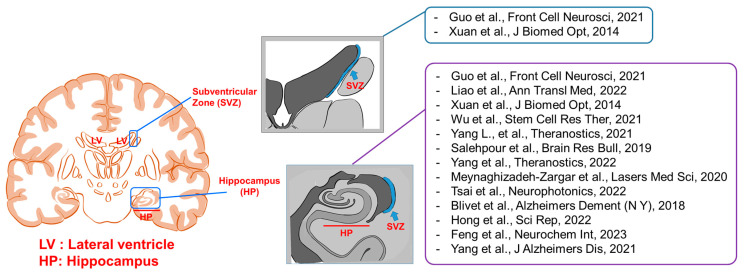
Anatomical targets for neurogenesis through photobiomodulation [35,36,37,38,39,40,41,42,43,44,45,46,47].

**Figure 2 ijms-24-15427-f002:**
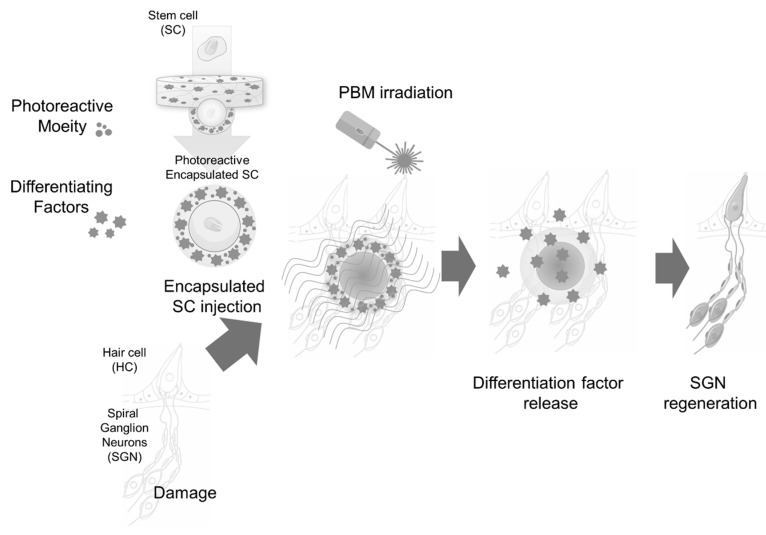
Schematic diagram of cochlear delivery of stem cells encapsulated in photoactive hydrogels with homing and differentiation factors.

**Table 1 ijms-24-15427-t001:** Summary of publications on photobiomodulation and neurogenesis.

Reference	Year	Light Source	Wavelength (nm)	Energy	Location	Animal Model	Target	Mechanism	Efficiency
[48]	2018	Diode laser	808	120 s × 350 mW/cm^2^/day Total: 294 J for 7 days	Scalp	PT stroke rat	Neuroprogenitor cells in the peri-infarct cortical region	Increased NeuN, MAP2, spinophilin, synaptophysin, BrdU, Ki67, and DCX Increased mitochondrial Cytc oxidase activity and ATP levels Decreased GPAP and Iba-1 Decreased TNF-α, IL-6, and IL-18; increased IL-4 and IL-10 Decreased CD32, 86, and iNOS; increased ARG1, TGFβ1, and CD206	Improved cortical neurogenesis Enhanced mitochondrial function Suppresses reactive gliosis Inhibited pro-inflammatory cytokines
[35]	2021	Diode laser	808	120 s × 20 mW/cm^2^ Total: 7.2 J/cm^2^ for 3 days	Cortical surface	Global cerebral ischemia rat	Hippocampal astrocyte and microglia cells	Increased BrdU and DCX-NeuN Decreased GFAP, NLRP3, cleaved IL-1β, and Iba1	Improved cognitive function Elevated endogenous neural stem cell proliferation Protected reactive astrocytes and microglia cells at 58 days
[36]	2022	Helium–neon laser	660	600 s × 5 mW/cm^2^ Total: 42 J/cm^2^ for 14 days	Scalp and skull	HI brain damage rat	Hippocampal neural stem cells	Increased Edu and Nestin Increased p-GSK3β, β-catenin, cyclin D1, PI3K, and p-AKT,	Enhanced spatial learning and memory Promoted hippocampal neural stem cell proliferation
[37]	2014	Laser	810	720 s × 25 mW/cm^2^ Total: 54 J/cm^2^ for 3 days	Transcranial	CCI–TBI injury mouse	Hippocampal and SVZ neurons	Increased BrdU, DCX, and TUJ1 Downregulated caspase 3	Enhanced learning and memory Increased neurogenesis in hippocampus and subventricular zone
[38]	2021	Semiconductor laser	635	600 s × 0.1 mW/cm^2^ Total: 60 J/cm^2^ for 1 month	Scalp and skull	Amyloid precursor protein/presenilin 1 transgenic mouse	Hippocampal neural stem cells	Increased DCX, Nestin, and TUJ1; decreased GFAP Activated TGFβ1 Increased Smad2/3 with Smad4; decreased Smad1/5/9 with Smad4	Enhanced spatial learning and memory Improved adult hippocampal neurogenesis

ATP, adenosine triphosphate; BrdU, bromodeoxyuridine; CCI, controlled cortical impact; Cytc, cytochrome c; DCX, doublecortin; HI, hypoxic–ischemic; p-GSK3β, glycogen synthase kinase 3β; IL, interleukin; PT, photothrombotic; SVZ, subventricular zone; TBI, traumatic brain injury; TGFβ1, transforming growth factor β1.

**Table 2 ijms-24-15427-t002:** Summary of publications on photobiomodulation and neuroprotective effects, glial cells, and synaptogenesis.

Reference	Year	Light Source	Wavelength (nm)		Energy	Anatomical Location	Animal Model	Target Cells	Mechanism	Efficiency
[49]	2019	Diode laser	808		36 s × 8 mW/cm^2^ Total: 4 J/cm^2^ for 14 days	Scalp	Global cerebral ischemia rat	Hippocampal CA1 pyramidal neuron cells	Increased NeuN, Tom20, Opa1, Mfn1, MMP, and FD Downregulated Bax/Bcl2, Cytc, Apaf1-casp-9, caspase 3, Annexin V, Drip1 GTPase, and Mdivi-1	Enhanced spatial learning and memory Protected hippocampal CA1 neurons Inhibited mitochondrial fragmentation and promoted mitochondrial function
[39]	2021	Diode laser	808		120 s × 8 mW/cm^2^ Total: 2.88 J/cm^2^/week for 3 weeks	Abdomen of pregnant rat from gestation day 1 to 21	Neonatal HI rat	Myeloid cells and hippocampal astrocytes	Increased synaptophysin, spinophilin, MAP2, MBP, NeuN, ATP, mfn2, MitoRed, iNOS, CD32, and S100 Decreased F-Jade C, caspase 3, caspase 9, Iba-1, GFAP, and C3d	Promoted synaptogenesis Increased memory and mitochondrial function Reduced inflammation and oxidative stress
[40]	2019	GaAlAs laser	810		5 s × 6.66 W/cm^2^ Total: 166.5 J/cm^2^ for 5 days	Transcranial Scalp	Sub-chronic restraint stress mouse	Prefrontal cortex and hippocampal neuron cells	Increased TAC, GSH, GPx, and SOD Decreased NF-kB, p38, and JNK Downregulated BAX/Bcl2, Cytc, and caspase-3 and −9	Ameliorated depressive-like behaviors Suppressed neuroinflammation, astrogliosis, and microgliosis
[41]	2022	Laser	808		120 s × 350 mW/cm^2^ Total: 126 J/cm^2^/week for 16 months	Scalp	Transgenic TgF344-AD rat	Microglial cells and astrocytes in cortex and hippocampus	Increased MBP, MAP2, NeuN, PSD, PSD95, Spin, and SYP Increased bouton vesicles and ATP Increased NeuN, Ki67, BrdU, and DCX	Improved typical AD pathology Recruited microglia surrounding amyloid plaques
[42]	2020	GaAlAs diode laser	810		1.7 s × 4.75 W/cm^2^ Total: 5 J/cm^2^/week for 4 weeks	Transcranial	Unpredictable chronic mild stress mouse	Hippocampal cells	Decreased NO, ROS and SOD activity, and serum cortisol Increased TAC and glutathione peroxidase	Improved anxiety-like behavior Promoted spatial learning and memory Reduced inflammation and oxidative stress
[43]	2022	GaAlAs diode laser	808		100 s × 1.333 W/cm^2^ Total: 133.3 J/cm^2^	Transcranial	Pentylenetetrazole-induced SE rat	Hippocampal microglial cells and astrocytes	Decreased NSE and GFAP Decreased Iba-1 in CA3 Increased MT-CO1 in CA3	Suppressed neuroinflammation, astrogliosis, and microgliosis
[50]	2021	Diode laser	780		120 s × 0.083 W/cm^2^ Total: 30 J/cm^2^/week for 60 days	Transcranial	Ischemic stroke rat	Brain tissue cells in the peri-lesional region	Increased GFAP; decreased Iba Decreased TNF-α, IL-1β, and IL6	Increased astroglial activity Attenuated neuroinflammation
[44]	2018	RGn500 (LED + laser) with static magnetic field	625 850	LED	600 s × 28 mW/cm^2^ Total: 117.6 J/cm^2^ for 7 days	Transcranial and abdomen	Aβ_25-35_ peptide toxicity mouse	Hippocampal and frontal cortex cells	Decreased lipid peroxidation Decreased GFAP, Iba1, TNF-α, IL-1β, and IL-6 Downregulated BAX/Bcl2 Decreased Aβ1-42 and pTau	Enhanced memory restoration Inhibited oxidative stress and intrinsic apoptosis Alleviated AD phenotype
			850	Laser						
[45]	2022	Diode laser	830		750 s × 30 mW/cm^2^ Total: 180 J/cm^2^	Scalp	SE mouse	Maturing granular cells in the hilus	Increased NeuN, Ki67, BrdU, and DCX	Increased cell proliferation and migration Increased CA1 pyramidal neurons
[46]	2023	Diode laser	808		120 s × 350 mW/cm^2^ Total: 294 J/cm^2^ for 7 days	Skull	PT stroke rat	Neuronal dendrites and cortex astrocytes	Increased MAP2, synaptophsin, and S100A10 Inhibited GFAP and C3d Decreased caspase 9 and BAX Increased Bcl-xL	Inhibited neurotoxic astrocytic polarization Preserved synaptic integrity
[47]	2021	Diode laser	808		120 s × 350 mW/cm^2^ Total: 126 J/cm^2^/week for 8 months	Scalp	Transgenic TgF344-AD disease rat		Increased MAP2, PSD95, Cytc oxidase activity, ATP, IL-4, and IL-10 Decreased caspase 3-, 9, F-jade C, Iba-1, NF-κB, TNF-α, and IL-1β Increased SOD2 activity; decreased MDA, 4-HNE, and p.H2A.X	Improved spatial memory Attenuated anxiety- and depression-like behaviors Inhibited neuroinflammation and oxidative stress

Aβ_25–35_, amyloid β_25–35_; AD, Alzheimer’s disease; ATP, adenosine triphosphate; BrdU, bromodeoxyuridine; Cytc, cytochrome c; DCX, doublecortin; GaAlAs, gallium aluminum arsenide; HI, hypoxic–ischemic; IL, interleukin; LED, light-emitting diode; NF-κB, nuclear factor kappa B; PT, photothrombotic; SE, status epilepticus; TNF-α, tumor necrosis factor-α.

## Data Availability

No new data were created or analyzed in this study. Data sharing is not applicable to this article.

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
