# Peer review of "Photobiomodulation of Neurogenesis through the Enhancement of Stem Cell and Neural Progenitor Differentiation in the Central and Peripheral Nervous Systems"

_ijms, 2023, doi:10.3390/ijms242015427_

Round 1
Reviewer 1 Report
The review article is devoted to describing the current state of affairs in the field of stimulating the restoration of nerve cells using photobiomodulation (red (600–700 nm) and NIR (700–1100 nm) regions). The mechanisms of stimulation of nerve cells are described.
The article discusses the effects of photobiomodulation on Neurogenesis, Neuroprotective effects of photobiomodulation, Effects of photobiomodulation on peripheral sensory neural structures, Hearing loss and others. Thus, the authors conducted an in-depth analysis of the problem.
In conclusion, they indicated that the development of treatment methods using photobiomodulation requires research on parameters such as the wavelength, energy density, treatment duration, and timing of PBM application. In addition, it is important to standardize experimental conditions and consider variability in cell responses.
There are several comments to the article:
The authors’ statement: “Nevertheless, neural structures are still thought to have weak regenerative capacity compared to other systems” remained without references to literary sources. I think it is necessary to provide links to articles here.
In my opinion, the phrase: “An overview of the potential targets for PBM-based neurogenesis is shown in Figure 1.” should end with square brackets containing the numbers of references to the literary sources indicated in the figure. Accordingly, all these references should be included in the list of references at the end of the article under the corresponding numbers.
I believe that after making these changes, the article can be accepted for publication.
Reviewer 2 Report
In this review article, the authors reviewed and discussed the effects of photobiomodulation (PBM) on neurogenesis in the CNS and the PNS to illuminate its potential as a therapeutic tool for neural degenerative disorders.
Comments
The reviewer has some concerns as follows:
1. The aims of this review article need to be revised. In lines 22-25 and 76-78, the description of the aims is not quite correct. This manuscript is a review article rather than an original research article.
2. In lines 54-57, the description for “PBM research is being conducted worldwide……in various medical applications including sports medicine” needs the references to support.
3. In Figure 1, please add the number of the cited literature shown in the reference list on the figure to facilitate readers' search. Moreover, the abbreviation of “PBM” shown in the figure legend is redundant and can be deleted.
4. The presentation of Figure 2 is confusing and unconvincing. The figure is a bit complicated and incomprehensible. What does the needle on image of cochlea mean? Is it the same as the first needle on animal ear? What are the red star-shaped dots on the image in the middle of the figure? Please simplify and make this diagram more precise.
5. The possible issues of safety or side effect for PBM application, which have been mentioned in the literature, can be described.
Round 2
Reviewer 2 Report
This revised manuscript can be accepted. No further comments.